# Characterization of commercial poultry farms in Mexico: Towards a better understanding of biosecurity practices and antibiotic usage patterns

Erika Ornelas-Eusebio[1,2,3], Gary García-Espinosa[2], Karine Laroucau[3], Gina Zanella[1]*

1 Epidemiology Unit, Laboratory for Animal Health, ANSES, University Paris-Est, Maisons-Alfort, France, 2 Department of Avian Medicine and Poultry Husbandry, Faculty of Veterinary Medicine and Animal Production, National Autonomous University of Mexico, Coyoacan, CDMX, Mexico, 3 Bacterial Zoonosis Unit, Laboratory for Animal Health, ANSES, University Paris-Est, Maisons-Alfort, France

* gina.zanella@anses.fr

## Abstract

Mexico is one of the world's major poultry producing countries. Two significant challenges currently facing the poultry industry are the responsible and judicious use of antimicrobials, and the potential occurrence of infectious disease outbreaks. For example, repeated outbreaks of highly pathogenic avian influenza virus subtype H7N3 have occurred in poultry since its first detection in Mexico in 2012. Both of these challenges can be addressed through good husbandry practices and the application of on-farm biosecurity measures. The aims of this study were: (i) to assess the biosecurity measures practiced across different types of poultry farms in Mexico, and (ii) to collect information regarding antimicrobial usage. A cross-sectional study was carried out through on-farm interviews on 43 poultry farms. A multiple correspondence analysis was performed to characterize the farms based on their pattern of biosecurity practices and antimicrobial usage. Three clusters of farms were identified using an agglomerative hierarchical cluster analysis. In each cluster, a specific farm type was predominant. The biosecurity measures that significantly differentiated the visited farms, thus allowing their clusterization, were: the use of personal protective equipment (e.g. face masks, hair caps, and eye protection), the requirement for a hygiene protocol before and after entering the farm, the use of exclusive working clothes by staff and visitors, footbath presence at the barn entrance, and the mortality disposal strategy. The more stringent the biosecurity measures on farms within a cluster, the fewer the farms that used antimicrobials. Farms with more biosecurity breaches used antimicrobials considered critically important for public health. These findings could be helpful to understand how to guide strategies to reinforce compliance with biosecurity practices identified as critical according to the farm type. We conclude by providing certain recommendations to improve on-farm biosecurity measures.

**Data Availability Statement:** All relevant data are within the manuscript and its Supporting Information files.

**Funding:** EOE acknowledges the doctoral scholarship No.600997/337987 received from the Consejo Nacional de Ciencia y Tecnología (CONACyT; https://www.conacyt.gob.mx/). All co-authors acknowledge the Mexico-France ECOS Nord funding program grant M17A01-ECOS Nord /291241 CONACyT- ANUIES. The funder had no role in study design, data collection and analysis, decision to publish, or preparation of the manuscript.

**Competing interests:** The authors have declared that no competing interests exist.

# Introduction

Biosecurity is the set of practices implemented with the objective of preventing the introduction and dissemination of infectious agents in an animal population [1], but also to prevent potential zoonoses [2]. It has been extensively demonstrated for poultry farms that implementing proper biosecurity practices contributes not only to the control of pathogen exposure [3–7], but also to improved productive performance [4, 8, 9], as well as to reduced antimicrobial usage [10, 11].

The misuse and overuse of antimicrobials have been linked to the rise of antimicrobial-resistant bacteria [12–14]. It is estimated that more than 70% of all antimicrobials sold worldwide are used in food-producing animals [12]. In particular, the poultry industry has been associated with regular use of antimicrobials [15–17]. However, since a clear association has been established between the extent of antimicrobial use (AMU) in livestock and the development of antimicrobial resistance (AMR) [18–20], increased awareness is pushing the poultry sector towards reduced and rational use of antimicrobials [13]. The World Health Organization (WHO) recommends monitoring of AMU in food-producing animals to protect human health. However, information about AMU and AMR in livestock is still poorly documented in low– and middle–income countries, mostly due to an absence of systematic surveillance systems [12, 14, 15, 21]. As a response to this issue, the Mexican government officialized the National Strategy to fight antimicrobial resistance under the One Health approach in 2018 [22], recommending that veterinary practitioners use antimicrobial compounds in food-producing animals more consciously, as well as to generate more information about AMR. At the same time, the government issued manuals on good husbandry practices for broiler chickens and laying hens, banning the use of antibiotics for preventive purposes or as growth-promoters, and recommending biosecurity measures that farms should impelement [23, 24]. Even though the guidelines established in these manuals are not mandatory, an increasing number of farms are aiming to certify their procedures according to the quality criteria of good husbandry practices, AMU included. Furthermore, the World Organization for Animal Health (OIE) has developed a list of critically important antimicrobial agents in veterinary medicine to help with the establishment of official national policies to assist veterinary practitioners in their therapeutic choices [25]. Only some antimicrobials have been reserved for food-producing animals, and they have been classified according to their importance as treatment for animal diseases, and their essential role to treat specific infections when there are no other therapeutic alternatives [25]. In Mexico, there is a high rate of antimicrobial resistance both in the community and as hospital-acquired infections; moreover, the few antimicrobial susceptibility studies conducted on foodborne and veterinary relevant pathogens have shown that antimicrobial resistance is an urgent problem to address [14].

Mexico is one of the largest poultry producing countries in the world. In 2019, it was the 4th raking country in egg production and ranked 6th in chicken meat production worldwide [26]. For the same year, the Mexican commercial poultry industry represented 64% of national livestock, with only a few companies sharing most of the market [27]. The sector is composed predominantly of highly integrated large-scale companies, with high standards of health and biosecurity practices, and high productive performance [28, 29]. The national performance of the poultry industry in Mexico is similar to that of the United States, but with a higher mortality rate due to the occurrence of diseases [28].

In 2012, a highly pathogenic avian influenza (HPAI) virus subtype H7N3 was reported for the first time on commercial poultry farms in Mexico [30]. The first outbreak, detected in laying hen flocks in a region with high poultry density, spread within a few months to broilers, breeders and backyard flocks [31]. Importantly, two poultry workers were found to be infected

with this virus [32]. The 2012 outbreak represented not only an animal health emergency but also a poultry production crisis, resulting in the death of over 22 million birds (either infected or culled), and an economic impact estimated at USD 720 million [30, 33]. This H7N3 subtype has been circulating in poultry in Mexico since its first detection, causing repeated outbreaks both in commercial chickens and in backyard poultry [33–36]. Moreover, it has been suggested that Mexico could play a potential role as a hotspot for viral interchange, as it is located along various winter migration routes of wild birds [31].

Compliance with biosecurity measures on poultry farms is often variable [16, 37]. This could be explained by several factors including the lack of training programs for staff, poor awareness of the potential consequences of a breach, lack of incentives for workers, and negative attitudes towards biosecurity, among others [37–39]. The effectiveness of biosecurity measures relies on their consistent application by all actors involved in poultry farming: field veterinarians, technicians, poultry workers, and visitors. Additionally, biosecurity practices should always be supported by the implementation of good husbandry practices [40].

The implementation of the formerly mentioned regulations by the Mexican government, coupled with the recent occurrence of HPAI outbreaks in poultry, encouraged us to perform this study aiming (i) to assess the biosecurity measures practiced across different types of poultry farms in Mexico, and (ii) to collect information regarding antimicrobial usage in different poultry farm types, using multivariate data analysis.

## Materials and methods

### Farm selection

Veterinarians who provide technical support to farms located in the main Mexican poultry producing states were invited to participate in the study during a national congress on poultry farming in 2017, there were no refusals. A total of 43 large-scale commercial farms were included in the study. Minimum sample size was calculated for a prevalence and risk factor study [41], and the maximum of visited farms was established considering time and resources available. We sought to include farms with different degrees of confinement (controlled environment *vs* open-sided houses), and farms raising chicken broilers and laying hens. The controlled environment houses are closed barns with airflow provided by an automatic tunnel ventilation system and artificial lighting, while open-sided houses are barns with open walls allowing natural ventilation, modulated by manually-operated curtains.

### Data collection

A questionnaire consisting of 48 fill-in-the-blank and close-ended questions was designed to gather information regarding: (i) farm characteristics, (ii) farm and poultry management practices, (iii) cleaning and disinfection procedures, and (iv) biosecurity measures. The questionnaire was designed in view of the manuals for good husbandry practices for chicken broilers and laying hens issued by the Mexican government.

The on-farm interviews were conducted by the same interviewer between June 2017 and June 2018. Taking into account the distances between farms, and to avoid potential pathogen introduction or dispersion, a maximum of two farms were visited per day. Additionally, the interviewer followed the biosecurity protocol implemented for visitors by the majority of the establishments. Visits were performed wearing clean clothes (most of the time provided on the farm, otherwise, a single-use coverall was worn). Systematic hand sanitization and showering, when feasible, were conducted before and after entering the farm. Interviews were conducted either with the farm manager or the veterinarian in charge of poultry health.

The last sections of the questionnaire (i.e. farm and poultry management practices, cleaning and disinfection procedures, and biosecurity measures) were open questions, allowing the respondent to give a detailed answer, especially concerning antibiotic usage.

Interviewees provided verbal consent before the interview was started. The research protocol, which also included sampling of birds for a microbiological study that has already been published [41], was approved by the Institutional Subcommittee for the Care and Use of Experimental Animals of the National Autonomous University of Mexico (registration number DC-2018/1-4). This subcommittee ensures ethical handling of animals, as well as confidentiality and data protection of the information gathered through the questionnaires.

## Data analysis

Data gathered through the questionnaires were entered into a Microsoft Excel© datasheet. The 48-questions resulted in 50 variables. Four quantitative variables (number of birds per barn, number of barns per farm, number of workers per farm, and duration of the vacancy period) were transformed into qualitative variables. Categorical boundaries were established taking into account the quantiles as cut-off points. Number of birds per barn was divided into two categories ($\leq$ 22,000 and $>$ 22,000). Number of barns per farm was used to classify farms into small farms ($\leq$6 barns) and large farms ($>$6 barns). Number of workers per farm was divided into two categories ($\leq$ 3 and $>$ 3), as well as the duration of the vacancy period before restocking ($\leq$ 1 week and $>$ 1 week).

A multivariate analysis of the data collected through the interview-questionnaire process was performed with R, version 3.6.2 (R Core Team 3.6.2, 2019). A multiple correspondence analysis (MCA) was performed to summarize and visualize the multidimensional dataset constructed with individuals (i.e. farms) and the categorical variables describing them. MCA is the correspondence analysis of the indicator matrix, where the rows are the respondents and the columns are the dummy variables for each of the categories of the variables. The goals of this analysis are first to study the similarities between the individuals, then to study the relationships between the variables, while assessing the associations between each of the variable categories, in order to finally associate the study of the individuals with that of the variables, with the aim of characterizing the individuals through their pattern of variables [42]. In this way, the most important variables that contribute to explain the variations in the dataset are revealed. The cloud of individuals and variables is represented in a low-dimensional Euclidian space by maximizing the variance (inertia) of the projected cloud of points [42]. Inertia is a measure of variance, showing the dispersion of data around their center of gravity, i.e. the dispersion of individual profiles around the average profile. In addition, eigenvalues are computed, which represent the contribution of each dimension to the total inertia, with the highest eigenvalue in the first dimension, and decreasing gradually for the rest of the dimensions. The eigenvalue is used to select the maximum number of dimensions to be included in the MCA–a value $\leq$ 0.5 is not usually considered [43, 44]. Graphical representations of the distances between individuals and the links between variables and their categories are also obtained. The distance between each point in the Euclidean space accounts for the variance between the points; therefore, the larger the distance, the lower the association.

In addition to the default indicator matrix, a Burt table was computed. This is the matrix of all pairwise associations between variables, including the diagonal associations between each variable and itself [45]. In this table, only the information about the relationships between categories is present, and not the information about the individuals [42]. The advantage of the Burt table is that theoretical eigenvalues obtained from it provide a better approximation of the inertia explained by the dimensions, as they are the squares of those obtained through the

analysis of the indicator matrix. Although these values are theoretical, they yield the same coordinates for individuals and variable categories as the analysis performed from the indicator matrix.

Using the dimensions with the greatest variance (inertia) generated by the MCA, the farms were classified into clusters through an agglomerative hierarchical cluster analysis (HCA) based on Ward's method, which consists in adding two groups (clusters) such that the growth of within-group inertia is minimal at each step of the algorithm. The hierarchical clustering algorithm can be visualized using a dendrogram. Within-group inertia characterizes the homogeneity of a cluster [46, 47]. The FactoMineR package was used to perform the MCA and HCA, and the factoextra package was used to visualize the outputs [48, 49]. The optimal number of clusters was validated using the NbClust package that provides 30 indices for determining the number of clusters and proposes the best clustering scheme from the different results obtained [47, 50].

Data for constructing the poultry density map on commercial poultry farms per federal Mexican state were obtained from the Secretariat of Agriculture, Livestock, Rural Development, Fisheries and Food (SAGARPA) [51], and imported into QGIS version 3.8.3 (2020) [52] using different layers for it construction [53, 54].

## Results

### General description of the farms

The 43 farms included in the study were located in the seven states of Mexico where the highest poultry production is found at the national level, and therefore that have the most significant health challenges to address. These states were Chiapas (13 farms), Guanajuato (6 farms), Jalisco (3 farms), Mexico City (3 farms), Morelos (12 farms), Puebla (5 farms), and Veracruz (1 farm) (Fig 1). The farms were located in temperate to tropical wet areas, between latitudes 20°49′01″N to 16°45′11″N and longitudes 102°43′59″W to 93°06′56″W.

Visited farms represented the major housing systems in the Mexican poultry industry: farms with controlled environment barns (n = 15) and farms with open-sided barns (n = 28). All farms had a well-defined fenced perimeter with specific monitored access points and were restricted to authorized personnel. Feed mills were integrated by the major companies owning these poultry farms. Ten farms were specialized in laying hens and 33 farms in chicken broiler breeding. All chicken broiler farms used fast-growing breeds (mainly the Ross and Cobb genetic lines), and males and females were bred separately in all of them, except one. Laying hen genetic lines included predominantly Bovans and Hy-Line. All-in/all-out systems were systematically applied by barn on all farms. The farm purpose was exclusive. Breeding of other poultry species on the same farm was not reported. All laying hens were housed in battery cages and all chicken broilers in barns, with the floor covered with at least 2 to 5 cm of single use litter. The most frequent bedding materials used were sawdust, coffee husk and rice hulls, with 39.4% for each of the first two materials, and 21.2% for rice hulls.

Flock size per barn on the chicken broiler and laying hen farms ranged from 1,000 to 38,000 and from 2,000 to 150,000 birds, respectively. There were between 2 and 6 barns per farm on 61% (n = 20/33) of the chicken broiler farms, and in half (n = 5/10) of the laying hen farms. On 39% (n = 13/33) of the chicken broiler farms and on the other half of the laying hen farms, there were between 7 and 16 barns. Most of the chicken broiler farms (63.6%, n = 21/33) and 60% of the laying hen farms (n = 6/10) employed at least four workers. The number of workers employed by chicken broiler and laying hen farms ranged from 2 to 8 and from 2 to 12 workers, respectively.

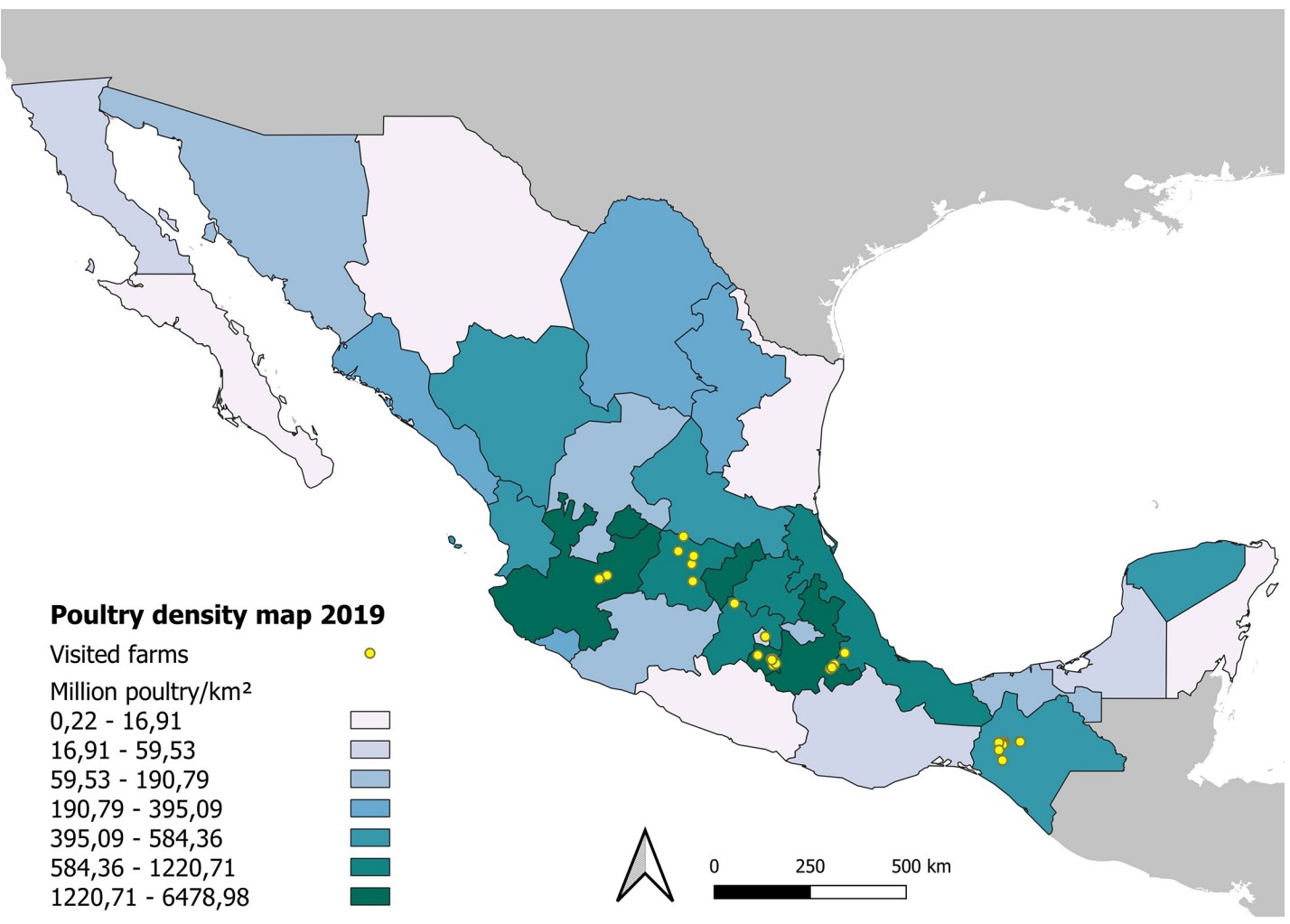

**Fig 1. Poultry density map per federal Mexican state with the location of the 43 poultry farms included in the study.**

## Multiple correspondence analysis

Of the 50 variables, 19 were retained for the MCA analysis following recommendations provided by Husson, et al. 2016 [46], seven of which described farm characteristics, six management practices, and five adopted on-farm biosecurity measures. The remaining 31 variables were dismissed for the following reasons: homogeneity in the response among the interviewees (13), their use to identify the farm and to describe its location (4), binary variables for which 5% or less of respondents gave the same answer (3), variables that were transformed into a new one (4), and the low pertinence of the obtained information (7). Farm type and farm purpose were introduced as supplementary (or illustrative) variables in the analysis, meaning that they had no influence on the dimension construction but they helped in result interpretation.

The MCA was performed keeping the first five dimensions covering 80.8% of the data variance with none of the remaining dimensions explaining more than 5% of the data variance (Table 1). Eigenvalues obtained from the Burt table showed that three dimensions already covered 89.1% of the data variance, while the rest of the dimensions explained <5%.

The variables more significantly related ($p<0.001$) to the construction of the first dimension were: (i) the mortality disposal strategy ($R^2 = 0.67$); (ii) the use of phosphonic acid derivatives

**Table 1. Eigenvalues and proportion of explained variance for the first ten dimensions obtained from the multiple correspondence analysis conducted for 43 Mexican commercial poultry farms.**

| | Indicator matrix | | | Burt matrix | | |
|---|---|---|---|---|---|---|
| | Eigenvalue | Variance (%) | Cumulative variance (%) | Eigenvalue | Variance (%) | Cumulative variance (%) |
| Dim 1 | 0.337 | 27.306 | 27.306 | 0.108 | 43.339 | 43.339 |
| Dim 2 | 0.284 | 22.964 | 50.270 | 0.085 | 34.141 | 77.480 |
| Dim 3 | 0.178 | 14.384 | 64.654 | 0.029 | 11.683 | 89.164 |
| Dim 4 | 0.115 | 9.344 | 73.999 | 0.012 | 4.897 | 94.061 |
| Dim 5 | 0.084 | 6.815 | 80.814 | 0.007 | 2.655 | 96.716 |
| Dim 6 | 0.062 | 4.998 | 85.812 | 0.003 | 1.383 | 98.099 |
| Dim 7 | 0.047 | 3.825 | 89.636 | 0.002 | 0.807 | 98.905 |
| Dim 8 | 0.032 | 2.615 | 92.251 | 0.001 | 0.470 | 99.376 |
| Dim 9 | 0.022 | 1.765 | 94.017 | 0.000 | 0.189 | 99.564 |
| Dim 10 | 0.021 | 1.700 | 95.717 | 0.000 | 0.167 | 99.731 |

Eigenvalues represent the contribution of each dimension to explain the total variability of the biosecurity practices and antimicrobial use considered in the analysis.

as antimicrobial treatment ($R^2 = 0.66$), and (iii) the use of exclusive working clothes by staff and visitors ($R^2 = 0.52$). For the second dimension, the variables more significantly related ($p < 0.001$) to its construction were: (i) the use of personal protective equipment by staff and visitors (e.g. face masks, hair caps, and eye protection) ($R^2 = 0.82$), (ii) staff and visitor hygiene protocol requirement before and after entering the farm ($R^2 = 0.53$), (iii) and the use of quinolones as antimicrobial treatment ($R^2 = 0.51$).

## Hierarchical cluster analysis

Taking into account the highest relative loss of within-group inertia, the consolidated partition of the hierarchical dendrogram evidenced three clusters (Fig 2). This number was validated through the simultaneous evaluation of 20 available indexes of the NbClust package. Three clusters were proposed by the majority of them by an objective "voting process".

The biosecurity practices most significantly linked with the cluster partition ($p < 0.001$) were: (i) use of personal protective equipment by staff and visitors (e.g. face masks, hair caps, and eye protection); (ii) compulsory staff and visitor hygiene protocol before and after entering the farm; (iii) staff and visitor use of exclusive working clothes, (iv) footbath presence at barn entrance, and (v) mortality disposal method. Other variables contributed to the characterization of each of the three clusters with *p*-values < 0.05.

The detailed biosecurity practices and farm characteristics observed by cluster are shown in Table 2. All 12 farms within cluster 1 raised chicken broilers in open-sided type barns and were classified in the smallest category of birds per barn ($\leq 22,000$ birds/barn) and barns per farm ($\leq 6$ barns/farm). Staff and visitors on these farms did not wear exclusive working clothes and used personal protective equipment only occasionally. The only mortality disposal system used on these farms was burial, and the vacancy period of poultry premises for hygiene and sanitation purposes was one week or even less. Eight farms within this cluster had neighboring commercial and backyard farms within a distance of 3 km or less, while four had no neighboring poultry farms. The use of footbaths at the entrance of each barn and the compulsory nature of the hygiene protocol before and after entering the poultry living area for staff and visitors were also significant characteristics describing all the farms within this cluster.

Cluster 2 was composed mainly of specialized chicken broiler farms (n = 17/18) in controlled-environment type barns (n = 14/18), and encompassed the majority of farms

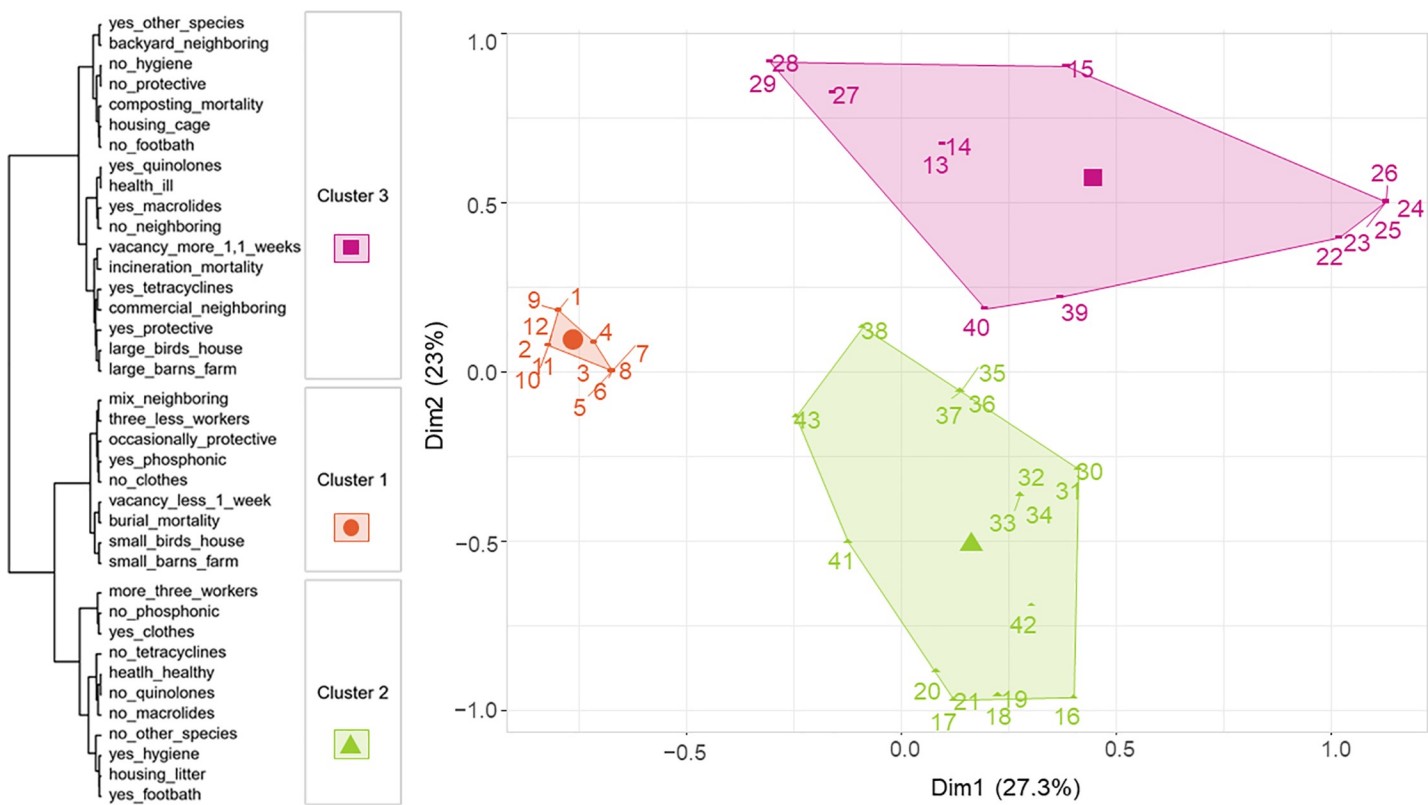

**Fig 2. Projection of the 43 Mexican commercial poultry farms included in the study within the three clusters identified through the HCA and plotted in the first two dimensions of the Euclidean space.** The dendrogram shows the categories of variables that most characterize the farms within each cluster.

categorized as the largest in terms of both number of barns per farm (n = 13/18), and number of birds per barn (n = 12/18). Staff and visitors at all farms within cluster 2 had to follow a mandatory hygiene protocol before and after entering the poultry living area, they were required to wear exclusive working clothes, and the presence of footbaths at the entrance of each barn was constant. The systematic use of personal protective equipment was reported on the majority of farms (n = 14/18). The primary mortality disposal method on farms within this cluster was incineration (n = 11/18), followed by burial (n = 7/18).

Cluster 3 mainly included farms with open-sided type barns (n = 12/13) specialized in laying hens (n = 9/13). The farm size was not significantly associated with this cluster; however, most of the farms (n = 7/13) were large in terms of number of birds per barn. On most of the farms, the staff were not required to follow a hygiene protocol to access the poultry living area (n = 12/13), nor to use personal protective equipment (n = 13/13). Likewise, there were no footbaths at the entrance of each barn in the majority of farms within this cluster (n = 9/13). The most significantly associated mortality disposal method was composting (n = 6/13) $p$-value < 0.001, even though burial (n = 3/13) was also significantly associated with farms in this cluster ($p<0.05$). The few studied farms on which the presence of other domestic species (cattle) was reported fell into cluster 3. A long barn vacancy period ($\geq$ 1 week) was significantly linked to farms within this cluster (n = 11/13); the maximum vacancy period reported was 22 days.

Among the variables with less than 5% variability or for which the answers were homogeneous, and therefore excluded from the MCA, it is worth mentioning the following: problems with rodent control mentioned only in one out of the 43 farms. On two farms, members of the

**Table 2. Frequency of biosecurity practices, antimicrobial usage and farm characteristics observed by cluster obtained from the multivariate analysis conducted on 43 commercial poultry farms in Mexico.**

| Variable | Category | Cluster 1 n = 12 farms (%) | | Cluster 2 n = 18 farms (%) | | Cluster 3 n = 13 farms (%) | | Overall n = 43 farms (%) |
|---|---|---|---|---|---|---|---|---|
| *Farm characteristics* | | | | | | | | |
| House type | Open-sided | **12 (100)** | ** | 4 (22) | *** | **12 (92)** | * | 28 (65) |
| | Controlled environment | 0 | ** | **14 (78)** | *** | 1 (8) | * | 15 (35) |
| Farm purpose | Broilers | **12 (100)** | * | **17 (94)** | * | 4 (31) | *** | 33 (77) |
| | Egg-laying hens | 0 | * | 1 (6) | * | **9 (69)** | *** | 10 (23) |
| No. of birds per barn | Small ($\leq$ 22,000) | **12 (100)** | *** | 6 (33) | * | 6 (46) | | 24 (56) |
| | Large ($>$ 22,000) | 0 | *** | **12 (67)** | * | 7 (54) | | 19 (44) |
| No. of barns per farm | Small ($\leq$ 6) | **12 (100)** | *** | 5 (28) | *** | 8 (62) | | 25 (58) |
| | Large ($>$ 6) | 0 | *** | **13 (72)** | *** | 5 (38) | | 18 (42) |
| No. of workers | $\leq$ 3 | 6 (50) | | 3 (17) | * | 7 (54) | | 16 (37) |
| | $>$ 3 | 6 (50) | | **15 (83)** | * | 6 (46) | | 27 (63) |
| Housing system | Litter | **12 (100)** | * | **18 (100)** | ** | 4 (31) | *** | 34 (79) |
| | Cage | 0 | * | 0 | ** | **9 (69)** | *** | 9 (21) |
| Neighboring farms < 3 km reported | Commercial | 0 | ** | **7 (39)** | | 6 (46) | | 13 (31) |
| | Backyard | 0 | | 1 (6) | | 3 (23) | | 4 (9) |
| | Both | **8 (67)** | * | 4 (22) | | 4 (31) | | 16 (37) |
| | None | 4 (33) | | 6 (33) | | **0** | * | 10 (23) |
| *Biosecurity measures* | | | | | | | | |
| Vacancy period | $\leq$ 1 week | **12 (100)** | *** | 9 (50) | | 2 (15) | ** | 23 (53) |
| | $>$ 1week | 0 | *** | 9 (50) | | **11 (85)** | ** | 20 (47) |
| Staff and visitor hygiene protocol before and after entering the farm | Compulsory | **12 (100)** | ** | **18 (100)** | *** | 1 (8) | *** | 31 (72) |
| | Optional or inexistent | 0 | ** | 0 | *** | **12 (92)** | *** | 12 (28) |
| Footbath at barn entrance | Yes | **12 (100)** | * | **18 (100)** | ** | 4 (31) | *** | 34 (79) |
| | No | 0 | * | 0 | ** | **9 (69)** | *** | 9 (21) |
| Use of exclusive farm clothes | Yes | 0 | *** | **18 (100)** | *** | 6 (46) | | 24 (56) |
| | No | **12 (100)** | *** | 0 | *** | 7 (54) | | 19 (44) |
| Personal protective equipment | Yes | 0 | ** | **14 (78)** | *** | 0 | ** | 14 (33) |
| | No | 0 | ** | 0 | *** | **13 (100)** | *** | 13 (30) |
| | Occasionally | **12 (100)** | *** | 4 (22) | | 0 | *** | 16 (37) |
| *Farm management practices* | | | | | | | | |
| Health status of the flocks | Healthy | 12 (100) | | **11 (61)** | ** | 12 (92) | | 35 (81) |
| | Ill | 0 | | 7 (39) | | 1 (8) | | 8 (19) |
| Breeding of other domestic species | Yes | 0 | | 0 | | **4 (31)** | ** | 4 (9) |
| | No | 12 (100) | | 18 (100) | | 9 (69) | ** | 39 (91) |

(*Continued*)

**Table 2.** (Continued)

| Variable | Category | Cluster 1 n = 12 farms (%) | | Cluster 2 n = 18 farms (%) | | Cluster 3 n = 13 farms (%) | | Overall n = 43 farms (%) |
|---|---|---|---|---|---|---|---|---|
| Mortality disposal | Burial | **12 (100)** | *** | 7 (39) | | 3 (23) | * | 22 (51) |
| | Incineration | 0 | ** | **11 (61)** | ** | 4 (31) | | 15 (35) |
| | Composting | 0 | | 0 | * | **6 (46)** | *** | 6 (14) |
| *Antimicrobial usage* | | | | | | | | |
| Phosphonic acid derivatives | Yes | **12 (100)** | *** | **0** | *** | 3 (23) | | 15 (35) |
| | No | 0 | | 18 (100) | | 10 (77) | | 28 (65) |
| Tetracyclines | Yes | **0** | ** | 6 (33) | | **7 (54)** | * | 13 (30) |
| | No | 12 (100) | | 12 (67) | | 6 (46) | | 30 (70) |
| Macrolides | Yes | 3 (25) | | 6 (33) | | 2 (15) | | 11 (36) |
| | No | 9 (75) | | 12 (67) | | 11 (85) | | 32 (74) |
| Quinolones | Yes | **0** | * | **8 (44)** | ** | 1 (8) | | 9 (21) |
| | No | 12 (100) | | 10 (56) | | 12 (92) | | 34 (79) |

Percentages indicate the proportion of farms included in the study representing this category and were grouped into each cluster. Significance of the link between the variable category and the cluster is expressed according to *p*-values (* *p*-value < 0.05; ** *p*-value < 0.01; *** *p*-value < 0.001). Categories that stood out within each cluster are highlighted in bold.

staff stated that they were not aware of potential zoonotic diseases associated with poultry, while training on continuous biosecurity and poultry disease prevention for staff was common across the rest of the analyzed farms; a hand washing facility and/or hand sanitizer availability at the entrance of each barn was absent on almost all farms (n = 41/43); however, handwashing facilities were in the clean room at the general entrance to each farm. Exhaustive cleaning and disinfection procedures during the vacancy period were reported on all farms. Several disinfectant products were mentioned, with organic acids the most extensively used.

## Pattern of antimicrobial usage

The use of antimicrobials as growth promoters was not reported on any farm. AMU was more extensive within farms belonging to clusters 1 and 3, with 100% (n = 12/12) and 85% (n = 11/13), respectively, while only 45% (n = 8/18) of farms from cluster 2 reported its usage. Four antimicrobial classes were reported to be used on the farms (Table 2), in decreasing order: phosphonic acid derivatives (n = 15/31), tetracyclines (n = 13/31), macrolides (n = 11/31), and quinolones (n = 9/31). On some farms, the use of more than 1 antimicrobial class was reported: 3 in cluster 1, 6 in cluster 2, and 1 in cluster 3.

Either the usage or the non-usage of certain antimicrobial classes on the farms was significantly associated with farms within each cluster. The prevailing antimicrobial used on all farms within cluster 1 (n = 12/12) was a phosphonic acid derivative (fosfomycin). In addition, three of these farms also reported the use of tylosin, a macrolide antibiotic. Thus, the potential interaction of fosfomycin and tylosin in flocks on these 3 farms was possible. The lack of use of tetracyclines and quinolones as antibiotics was significantly associated with the farms belonging to cluster 1. Quinolones were significantly associated with farms using antimicrobials (n = 8) within cluster 2. On six of these farms, tetracyclines and macrolides were also given. Thus, the potential interaction of tetracyclines, quinolones and macrolides in flocks within these 6 farms was possible. Conversely, the lack of use of phosphonic acid derivatives as

antimicrobials was significantly associated with farms belonging to this cluster. Tetracyclines were the antimicrobial class reported to be used on 7 farms within cluster 3, and the only antibiotic class whose use was significantly associated with them. Additionally, on 3 out of these 11 farms, a phosphonic acid derivative antimicrobial was used, while one used macrolides and one quinolones. On one out of the 11 farms belonging to cluster 3 that used antimicrobials, the use of tetracyclines and quinolones was reported; thus, the interaction of these antimicrobials in flocks on this farm was possible.

## Discussion

Significant variations in the application of biosecurity practices were observed across the farm clusters identified in our study. This finding is consistent with the results of previous studies showing that the on-farm application of biosecurity measures tends to be variable and could often be intermittent [37, 55], both on chicken broiler farms [3, 9, 56] and laying hen farms [57–59]. We conducted a multidimensional exploratory analysis considering that evaluation of biosecurity practices is measured by a large number of variables. As many of these variables may be correlated, this methodology makes it possible to uncover the relationships among categorical variables within and between farms, to ultimately find patterns [44, 46]. This information may not otherwise be discovered through a pairwise analysis [44]. The subsequent hierarchical clustering analysis conducted allowed us to objectively group the farms according to these previously identified patterns. This approach was adopted instead of describing the biosecurity practices implemented through specific farm characteristics, such as degree of confinement (open-sided barns *vs* controlled-environment barns), farm size or farm purpose (broilers *vs* layers).

Five biosecurity practices were identified as the most significantly associated with farm classification into three clusters. Three of these practices were related to measures concerning directly the staff or visitors (appropriate use of personal protective equipment, hygiene protocol before and after entering the farm, use of exclusive working clothes), while the last two were related to general farm facilities (i.e. footbath presence at barn entrance) and poultry mortality disposal methods. Previous studies have established that the implementation of and compliance with biosecurity measures regarding personnel are crucial to prevent the transmission of pathogens into a flock [7, 9, 37, 56, 59, 60]. In a study performed on poultry farms in the Netherlands, it was found that non-adherence by personnel to the hygiene protocols, and not wearing exclusive working clothes before entering the poultry living area, represented the highest transmission pathways of pathogens for poultry from an external source [59]. In fact, if staff or visitors have contact with infected birds and/or their feces and/or contaminated material, they could become the main source of contamination, within and between farms. The use of personal protective equipment is not intrinsically a biosecurity practice, but an occupational safety recommendation, as poultry workers are at increased risk of respiratory exposure to dust, particulate feathers, and atmospheric contaminants including ammonia and hydrogen sulfide during certain handling activities [61, 62]. Its usage is not mandatory for staff nor for visitors in accordance with Mexican law. However, according to Dorea et al., the mandatory usage of personal protective equipment, mainly for farm visitors, emerged as a response to address the threat of introduction of pathogens either by veterinarians who ensure technical support to the farms or by farmers [63]. In our study, the use of personal protective equipment was observed systematically on farms within cluster 2, and occasionally on some farms within cluster 1, the two clusters with better biosecurity practices, whereas its use was inexistent on farms within cluster 3. Furthermore, in a study performed in Latin-American poultry workers conducted in poultry processing plants in the United States, Arcury et al. found that the use of

personal protective equipment, coupled with receiving constant training on biosecurity, were significantly associated with a positive work safety climate, especially among employees in this sector [64].

Poultry farming faces a major environmental challenge associated with waste generation, its adequate treatment and disposal. As a result, methods considering both the environmental impact and safe waste disposal should be prioritized [65]. There are several methods and technologies for handling carcasses, each with its pros and cons. Burial is one of the least acceptable methods mainly due to environmental issues, e.g. the potential risk of ground water pollution due to adsorption of pollutants by the soil. Incineration is recognized as one of the biologically safest methods, while composting is becoming increasingly adopted as it has been successfully used for emergency disposal of carcasses [65, 66]. The only method for waste disposal used on farms from cluster 1 in our study was burial, which may represent a health risk. For instance, Tablante et al. found that farms on which carcasses were not properly buried–resulting in scavenging by other animals–experienced recurring infectious laryngotracheitis outbreaks [9]. In contrast, the majority of farms within cluster 2 opted for incineration. Even though its implementation is initially expensive and facility maintenance should be permanent, it is the safest disposal method as it does not attract scavengers or pests, and its residues can be safely disposed of without causing water quality problems [65]. The main method for carcass disposal on farms within cluster 3 was composting. When this is performed properly, pathogens are efficiently eliminated and the resulting material can be used in further agricultural processes [67, 68]. Farms within cluster 1 should improve their waste disposal methods, as in many cases they have enough space to perform composting; this could be an affordable, easy-to-implement solution.

It has been proposed that in general, most of the breaches in biosecurity practices are similar for laying hens and chicken broiler farms [59]. However, we did find a difference by using the clustering approach. Overall, farms from cluster 3, in which laying hen farms predominated, were more prone to breach biosecurity practices that had previously been identified as risk factors associated with low pathogenic outbreaks of influenza virus subtype H5N2 in laying hen farms in Japan [69]. These practices were the inexistence of or vague implementation of hygiene protocols before entering the poultry living area, no footbath at each barn entrance, and not using exclusive working clothes, thus coinciding with our findings. Interestingly, the first case of HPAI virus subtype H7N3 in Mexican poultry occurred on laying farms [30]. Moreover, the presence of animal species other than poultry e.g. cattle, observed only on farms within cluster 3, might be a relevant factor for avian influenza introduction and dissemination into the poultry premises, as shown in a previous study [59]. Cattle presence can generate additional personnel movements and activities related to cattle rearing (e.g. extra farm visits, feed-related activities) and, more importantly, they could have a potential role as pathogen carriers. Kalthoff et al. showed that cattle experimentally infected with an avian influenza virus can actually seroconvert and become asymptomatic shedders of the virus [70]. Conversely, more stringent and exhaustive biosecurity protocols were in place on farms from cluster 2, followed by farms from cluster 1, both clusters mostly encompassing chicken broiler farms. This finding is in agreement with the results of a study carried out by Scott et al. on Australian poultry farms, in which they observed that more demanding biosecurity measures were practiced on chicken broiler farms than on laying hen farms [71]. The authors of this study also found that footbaths where absent at each barn entrance on all laying hen farms, a breach that we also observed. It would be interesting to investigate the occurrence of avian influenza or/and other important poultry pathogen outbreaks on Mexican poultry farms to compare their frequencies according to the farm purpose.

Remarkably, in only 5% of the farms included in our study, a formal protocol for hand washing before and after entering each barn was described. This finding is in accordance with results of the study carried out by Racicot et al. (2011) on chicken broiler and laying hen farms, in which they found that one of the most frequent breaches by staff was related to hand sanitizing [37]. This is important because poorly sanitized hands can act as an efficient mechanism to spread pathogens within and between farms (Racicot *et al*. 2013). Furthermore, Racicot et al. (2011) also observed that waterless alcohol-based gel for hand sanitizing was better accepted by poultry workers [37]. However, its use should not replace hand washing with soap when visible organic material is present on hands (i.e. moderate to high contamination), because dirt significantly interferes with the microbicidal activity of handrubs [72, 73]. Several formulations and presentations are available for handrubs. Racicot et al. (2013) found that there was no difference in effectiveness between products and protocols only when the initial level of bacterial contamination was low; hence, prior hygiene of hands is essential in these cases [72]. Similarly, Wilkinson et al. found no difference regarding antibacterial efficacy attributable to isopropanol- (IPA) vs ethanol (EtOH)-based formulations [74]. However, in their study performed with 20 volunteers, EtOH-based handrubs in liquid or foam presentations were more comfortable for use because they dry faster than gel presentations (Wilkinson *et al*. 2018). This was further confirmed by Greenaway et al. who found that a 1.5 mL handrub dose yielded the most acceptable cost-effect result [75]. The WHO recommends the use of these alcohol-based handrubs in resource-limited or remote areas with lack of accessibility to sinks or other facilities for hand hygiene. This is a method for promoting hand hygiene compliance, by making the process faster and more convenient for workers [76]. Therefore, we propose the implementation of handrub dispensers at each barn entrance as an immediate alternative to correct this biosecurity breach.

No use of antimicrobials for growth promotion was reported on any farm, which is in alignment with national and international measures implemented to prevent antimicrobial resistance [77]. Four antimicrobial classes were reported to be used for treatment on 31 of the 43 visited farms: tetracyclines, quinolones, macrolides and phosphonic acid derivatives. According to the list of antimicrobial agents of veterinary importance issued by the OIE, the classes of antimicrobials used on the farms included in our study are approved for use in food-producing animals [25]. The WHO established a list of critically important antimicrobials for human medicine, whose scope is to classify those antimicrobials that are also used in veterinary medicine [78]. According to this list, of the 4 classes of antimicrobials used on the farms included in our study, tetracyclines are highly important for human medicine, and phosphonic acid derivatives are critically important, while quinolones and macrolides have the highest priority. WHO recommends that all antimicrobials should be used prudently in veterinary medicine, especially those classified as critically important and with the highest priority. In Mexico, there has been a list of antimicrobials allowed in veterinary medicine since 2012 [79]. However, this list does not classify the antimicrobial classes in relation to the risk to public health posed by their use in animals, leaving the therapeutic choice to the discretion of the veterinarian providing technical support to the farm. Conversely, the United States Food and Drug Administration (FDA) together with their veterinary authorities, have provided guidelines that include a list of antimicrobials approved for its use exclusively in poultry, and classified according to their importance in human medicine–these guidelines are also intended to aid veterinarians in their therapeutic decision-making [80]. In parallel, the European Medicines Agency (EMA) provides a categorization of antimicrobials for use in animals, reserving only some of them for use in food-producing animals [81]. Both guidelines, American and European, are in accordance with the list of critically important antimicrobials for human medicine established by the WHO. However, the classification scale of some antimicrobials could be more stringent on

either list in addressing their particular needs, e.g. the phosphonic acid derivative fosfomicyn is banned for use in veterinary medicine in Europe, and is not included in the antimicrobial schema for poultry in the United States.

The proportion of farms using antimicrobials differed by cluster; the more stringent the biosecurity measures on farms within a cluster, the fewer the farms that used antimicrobials. Specifically, antibiotics were used on only 45% of farms within cluster 2 *vs* 85% on farms within cluster 3, reaching even 100% on farms within cluster 1. In a study conducted on 60 German pig farms, Raasch et al. confirmed that the improvement of biosecurity measures is a feasible strategy to reduce antimicrobial usage at the herd level [8]. Similarly, Chauvin et al. observed that compliance with biosecurity practices was associated with a lower antimicrobial consumption level, after quantifying the consumption level of antibiotics in 246 turkey broiler flocks [10]. Furthermore, we found that the more breaches there were to on-farm biosecurity practices, the more likely it was to observe the use of antibiotics critical for human health. To illustrate this, fosfomycin was the most widely used antibiotic among farms reporting antimicrobial use in our study. Its use was extensive on farms within clusters 1 and 3, the two clusters of farms in which less stringent biosecurity measures were practiced, while the lack of its use was significantly associated with farms belonging to cluster 2. The antimicrobial class whose use was significantly associated with farms within cluster 2 was the tetracycline group, which is classified as highly but not critically important by the WHO. Fosfomycin is used to treat infections caused mainly by *E. coli* and *Salmonella* spp. in poultry, but to ensure its efficacy on susceptible bacteria, it must be used at specific concentrations under a specific schema [82, 83]. A 10-year longitudinal study of uropathogenic *E. coli* strains (UPEC) in humans in Mexico, identified these strains as the leading cause of urinary infections [84]. Moreover, rates of multidrug-resistant UPEC have significantly increased over time, reaching more than 60% of isolated strains, complicating their treatment, and leading to severe complications such as cystitis, pyelonephritis and urosepsis [85]. Fosfomycin is used mainly for the treatment of urinary tract infection in humans, with bacterial resistance arising readily *in vitro* [86]. In Mexico, fosfomycin represents the last-resort antimicrobial therapeutic alternative [87]. Therefore, we suggest to add to the Mexican manuals of good husbandry practices, a classification of the antimicrobial classes that are used in poultry aligned with WHO criteria. The aim would be to guide field veterinarians towards more judicious therapeutic choices and to restrict the usage of medically important antimicrobials only to specific situations. The use of such critical antimicrobials for humans in veterinary medicine is highly undesirable, especially in food-producing animals. This is because antimicrobial-resistant bacteria could develop in livestock and then spread to the environment through their feces or waste from processing plants. Human exposure to food or water contaminated by antimicrobial-resistant bacteria has been found to be the most common and efficient transmission route [13].

Few studies have been conducted on Mexican poultry farms assessing biosecurity practice implementation after the major 2012 HPAI outbreak [33, 88–90]. Following the first detection of an HPAI virus subtype H5N2 in Mexican commercial poultry in 1994 [91], the government initiated a national campaign for its control and eradication that has been maintained and updated considering good husbandry practices [92]. As part of this campaign, the government first issued in 2009, and updated in 2016, two manuals on good husbandry practices, one for chicken meat and the other for egg production in which health management, poultry nutrition, biosecurity measures, waste disposal and animal welfare, among other poultry farming practices, are described [23, 24]. The government's latest initiative to improve husbandry practices took place in June 2019 by presenting a protocol to strengthen biosecurity measures on commercial poultry farms that, once implemented, will be verified through official audits in

order to confirm that the farms meet the minimum biosecurity measures established in this protocol [93].

Our study has some limitations. Firstly, no random selection of the farms could be applied since no list is available of all commercial poultry farms in Mexico. Location of the farms (covering the most densely poultry populated states) and farm purpose (encompassing broiler and laying hen farms, the two most numerous nationwide farm purposes) were the two aspects we took into account to extend an invitation to a farm. Hence, we believe that the farms included in our study, which were already following the guidelines for the best practices on poultry husbandry [23, 24], may be representative of the homogeneous large-scale poultry farming sector in Mexico. Visiting several farms at different geographical locations over a fixed period of time could represent a biosecurity risk. For this reason, obtaining consent to visit and to perform the interviews on the commercial poultry farms was difficult. Nonetheless, there were no refusals. Finally, compliance with on-farm biosecurity measures could be questionable. Racicot et al. (2011) and Delpont et al. (2018) found that discrepancies between the implemented biosecurity measures and their actual practice are more frequent that one may expect, leading to a decrease in their effectiveness with the associated risks in terms of pathogen exposure and transmission [37, 94]. To take these possible discrepancies into account, our study design included on-farm visits and personal interviews to administer the questionnaire. We consider that this approach gave us the opportunity to gather complementary information through an open dialogue with the interviewees, with the understanding and reassurance that this was not an audit nor an official inspection, but an independent, anonymous study aiming to gather knowledge and assist the poultry sector. Only certain practices could be observed directly, but the bias of an external observer may have played a role. However, we assumed that since the studied farms belonged to large, well-integrated poultry companies, the implementation of and compliance with biosecurity measures would tend to be higher. In a study involving 921 Australian poultry farms, East (2007) showed that the implementation rates of biosecurity practices were higher in integrated companies than on independently owned farms [95]. In addition, a non-negligible number of variables representing the implementation of major biosecurity practices (16 out of 50) were dismissed from the analysis due to the lack of variability and homogeneity in the responses given by the interviewees. This fact can be interpreted as a positive consequence of the extensive implementation of these biosecurity practices on these poultry farms. For example, the existence of a perimeter fence, the implementation of a logbook, the use of the all-in/all-out system, the ban on breeding two poultry species or zootechnical purposes simultaneously in the same facilities, and the establishment of a vacancy period, have been implemented on all the visited farms. This is similar to the findings of East (2007), where the farms owned by a major company were compliant with all the major biosecurity practices evaluated [95]. In future studies, a scoring system could be used to overcome this homogeneity and more accurately assess the degree of compliance and not just the presence or absence of a given practice.

The certificate of best husbandry practices, issued by the Mexican government to farms that meet the guidelines established in the aforementioned manuals, is valid for one year, with the possibility of an audit to verify guideline compliance at any time. If a breach is detected during the validity period, the certificate could be canceled, depending on the severity of the fault. This encourages farms wishing to retain their certification to observe constant biosecurity practice compliance. We consider that audits, coupled with constant staff training and motivation, could positively influence compliance with biosecurity measures. In a study exploring the determinants of biosecurity practices on duck farms in France after an H5N8 highly pathogenic avian influenza epidemic, it was shown that farmers having better knowledge of and positive attitudes towards biosecurity had a significantly higher adoption of these measures [38].

Our study provides an exploratory analysis of patterns of on-farm biosecurity practices across the different groups of poultry farms in Mexico identified through our analysis. This could be helpful to field veterinarians or farmers to understand how to guide strategies to reinforce staff training, as well as for on-farm implementation and compliance, prioritizing the practices identified as critical in our analysis. This study also offers information characterizing antimicrobial use in the poultry industry, and thereby contributes to the national need for information on this subject. These data may help to consolidate a national strategy to improve the use of antimicrobials and contain antimicrobial resistance. We hope that our results could also be useful to other poultry industries with similar conditions outside Mexico. Further studies investigating the effectiveness of the official provisions issued in the last few years should be conducted, to follow up on trends in on-farm biosecurity practices and AMU in the Mexican poultry industry.

## Supporting information

**S1 QuestionnaireEN.**
(DOCX)

**S2 QuestionnaireES.**
(DOCX)

**S1 Data.**
(CSV)

**S2 Data.**
(R)

## Acknowledgments

We thank the poultry companies and veterinarians who kindly agreed to participate in the study. Special thanks to Dr. Rigoberto Hernández Castro for his valuable comments provided for the discussion section.

## Author Contributions

**Conceptualization:** Erika Ornelas-Eusebio, Gina Zanella.

**Data curation:** Erika Ornelas-Eusebio, Gina Zanella.

**Formal analysis:** Erika Ornelas-Eusebio, Gina Zanella.

**Funding acquisition:** Gary García-Espinosa, Karine Laroucau.

**Investigation:** Erika Ornelas-Eusebio, Gina Zanella.

**Methodology:** Erika Ornelas-Eusebio, Gina Zanella.

**Writing – review & editing:** Erika Ornelas-Eusebio, Gary García-Espinosa, Karine Laroucau, Gina Zanella.

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
