## [Decision Letter · Decision Letter 0]

3 Aug 2020

PONE-D-20-19059

Characterization of commercial poultry farms in Mexico: towards a better understanding of biosecurity practices and antibiotic usage patterns

PLOS ONE

Dear Dr. Ornelas Eusebio,

Thank you for submitting your manuscript to PLOS ONE. After careful consideration, we feel that it has merit but does not fully meet PLOS ONE’s publication criteria as it currently stands. Therefore, we invite you to submit a revised version of the manuscript that addresses the points raised during the review process.

Some clarifications regarding methodology and data presentation.

We look forward to receiving your revised manuscript.

Kind regards,

Iddya Karunasagar

Academic Editor

PLOS ONE

Journal Requirements:

3.We note that [Figure(s) 1] in your submission contain [map/satellite] images which may be copyrighted. All PLOS content is published under the Creative Commons Attribution License (CC BY 4.0), which means that the manuscript, images, and Supporting Information files will be freely available online, and any third party is permitted to access, download, copy, distribute, and use these materials in any way, even commercially, with proper attribution. For these reasons, we cannot publish previously copyrighted maps or satellite images created using proprietary data, such as Google software (Google Maps, Street View, and Earth). For more information, see our copyright guidelines: http://journals.plos.org/plosone/s/licenses-and-copyright.

1.    You may seek permission from the original copyright holder of Figure(s) [1] to publish the content specifically under the CC BY 4.0 license. 

Additional Editor Comments (if provided):

Two reviewers have commented on the manuscript. Clarifications are needed regarding sample size and basis of farm selection and other issues. Please address all comments point by point.

Reviewers' comments:

Reviewer's Responses to Questions

**Comments to the Author**

1. Is the manuscript technically sound, and do the data support the conclusions?

Reviewer #1: Yes

Reviewer #2: Partly

2. Has the statistical analysis been performed appropriately and rigorously? 

Reviewer #1: Yes

Reviewer #2: Yes

3. Have the authors made all data underlying the findings in their manuscript fully available?

Reviewer #1: Yes

Reviewer #2: No

4. Is the manuscript presented in an intelligible fashion and written in standard English?

Reviewer #1: Yes

Reviewer #2: Yes

5. Review Comments to the Author

Reviewer #1: This is a nice draft, which is presented in a way that audience could understand clearly. My specific suggestions are below:

Line 109: Please describe whether the farm selection was purposive or random within the mentioned categories/groups

Line 263: The caption of the Figure probably merged with another sentence which should be in the next paragraph, please check

Line 286: The caption of the Table probably merged with another sentence which should be in the next paragraph, please check

Line 290: Table 2 can be presented in landscape style, some % seems break between lines

Line 366: It seems that this should be the first paragraph of the discussion section as it contains the most important results of your paper

Line 532: The first paragraph of the discussion section can be placed here, before limitation

Line 579: This paragraph contains the same argument as the first paragraph of the discussion, seems repetition of idea

Reviewer #2: Please refer to the attached document with more detailed remarks. IN particular I am missing a justification of the sample size. Furthermore please discuss the robustness of MCA for n= 43 and 19 (or 50) variables.

6. PLOS authors have the option to publish the peer review history of their article (what does this mean?). If published, this will include your full peer review and any attached files.

Reviewer #1: **Yes: **Mahbub-Ul Alam

Reviewer #2: **Yes: **Sonja Hartnack

---

## [Author Response · Author response to Decision Letter 0]

22 Oct 2020

***Reviewer #1: 

This is a nice draft, which is presented in a way that audience could understand clearly. My specific suggestions are below:

Line 109: Please describe whether the farm selection was purposive or random within the mentioned categories/groups

Authors: In lines 111-113 it is mentioned that we contacted veterinarians who provide technical support to poultry farms during a national congress on poultry farming. When the invitation to participate in the study was extended, we corroborate with the veterinarians in order to encompass the mentioned categories, hence the selection was purposive (as specified in lines 115-117 “We sought to include farms…”). 

Line 263: The caption of the Figure probably merged with another sentence which should be in the next paragraph, please check.

Authors: The sentence “The dendrogram shows the categories of variables that most characterize the farms within each cluster”, corresponds to the legend of figure 2. 

Line 286: The caption of the Table probably merged with another sentence which should be in the next paragraph, please check.

Authors: It has been checked. The table has a legend after the title (lines 292-295).

Line 290: Table 2 can be presented in landscape style, some % seems break between lines.

Authors: The table 2 has been edited; percentages should appear in the same line. 

Line 366: It seems that this should be the first paragraph of the discussion section as it contains the most important results of your paper

Authors: Thank you for the suggestion, the change has been made.

Line 532: The first paragraph of the discussion section can be placed here, before limitation

Authors: Thank you for the suggestion, the change has been made.

Line 579: This paragraph contains the same argument as the first paragraph of the discussion, seems repetition of idea.

Authors: We agree, the last two ideas of the paragraph were removed, which were in fact a repetition of the last paragraph intended to conclude the article.

***Reviewer #2: 

Please refer to the attached document with more detailed remarks. In particular, I am missing a justification of the sample size. Furthermore, please discuss the robustness of MCA for n= 43 and 19 (or 50) variables.

PLoS endorsed already many years ago the STROBE guidelines. Please make sure that the reporting of this cross-sectional study complies with the guidelines: https://journals.plos.org/plosmedicine/article?id=10.1371/journal.pmed.0040297

Authors: We have checked the manuscript compliance according to the STROBE reporting guidelines. 

Please provide a justification for the sample size. What coverage of commercial farms do you expect by the 43 farms? 

Authors: The criteria used to establish the minimum and maximum of sample size was specified in the manuscript (lines 114-115). Minimum sample size was calculated for a prevalence and risk factor study that has already been published (10.1016/j.prevetmed.2020.104922.), and the maximum number of farms to visit was established considering time and resources available. We have specified this information in the manuscript (lines 114-115).

In line with the sample size, please discuss the robustness of your results for n=43 and 29 variables. I understood that 20 cases for each variable are suggested (https://doi.org/10.1007/s11135-015-0206-0). 

Authors: We were not aware of the Giovanni di Franco (2016) paper and his suggestion about a threshold of 20 individuals per active variable category. To conduct the MCA we followed the recommendations issued by Husson et al. (2016), who implemented the R FactoMineR package we used. Husson’s recommendations are in line with those preconized in di Franco’s paper (i.e. eliminate low-frequency categories, recoding the variables in order to rebalance the distribution of the cases within the categories of each variable, balance the number of categories per single variable). We also excluded variables with less than 5% variability for which the answers were homogeneous. Husson et al. do not preconize a minimal number of cases per active variable. We are aware that if we took into account this threshold, some categories in our study would be below it and therefore should be excluded, however, if the threshold is 15, there would be fewer variable excluded. If that rule of thumb was to be applied even with much more poultry farms included in our study, it would have been very difficult to have 20 cases in each active category, for instance: farms with no footbaths at barn entrance, farms breeding other domestic species, farms implementing composting (which is a recently implemented practice on poultry farms in Mexico). We have specified that we followed recommendations provided by Husson, et al. 2016 in the results section (line 233-234). 

How many veterinarians or farmers refused to participate?

Authors: There were no refusals, although, obtaining consent to visit the commercial poultry farms and to perform the on-farm interviews was difficult. We have specified it in line 113 and modified a passage in the discussion to take into account this fact (lines 541-544).

Did you perform any pre-testing of the questionnaire? 

Authors: Yes, we did. The final draft of the questionnaire was pre-tested on five veterinarians specialized in poultry science in order to identify ambiguities in the questions and their relevance. Minor suggestions were made by the respondents (e.g. formulate open-ended instead of multiple-choice questions and increase options in the multiple-choice questions). Their recommendations were taken into account.

L 105-107 Please formulate your research hypothesis in more precisely (e.g. similar to l. 25-27).

Authors: The hypothesis has been reworded as suggested (lines 105-108).

---

## [Decision Letter · Decision Letter 1]

2 Nov 2020

Characterization of commercial poultry farms in Mexico: towards a better understanding of biosecurity practices and antibiotic usage patterns

PONE-D-20-19059R1

Dear Dr. Ornelas Eusebio,

We’re pleased to inform you that your manuscript has been judged scientifically suitable for publication and will be formally accepted for publication once it meets all outstanding technical requirements.

Kind regards,

Iddya Karunasagar

Academic Editor

PLOS ONE

Additional Editor Comments (optional):

All reviewer comments have been addressed.

Reviewers' comments:

Reviewer's Responses to Questions

**Comments to the Author**

1. If the authors have adequately addressed your comments raised in a previous round of review and you feel that this manuscript is now acceptable for publication, you may indicate that here to bypass the “Comments to the Author” section, enter your conflict of interest statement in the “Confidential to Editor” section, and submit your "Accept" recommendation.

Reviewer #2: All comments have been addressed

2. Is the manuscript technically sound, and do the data support the conclusions?

Reviewer #2: Yes

3. Has the statistical analysis been performed appropriately and rigorously? 

Reviewer #2: Yes

4. Have the authors made all data underlying the findings in their manuscript fully available?

Reviewer #2: No

5. Is the manuscript presented in an intelligible fashion and written in standard English?

Reviewer #2: Yes

6. Review Comments to the Author

Reviewer #2: All my comments have been adressed. Please consider to make the data set as well as your annotaed R code available.

7. PLOS authors have the option to publish the peer review history of their article (what does this mean?). If published, this will include your full peer review and any attached files.

Reviewer #2: **Yes: **Sonja Hartnack

---

## [Editor Report · Acceptance letter]

16 Nov 2020

PONE-D-20-19059R1 

Characterization of commercial poultry farms in Mexico: towards a better understanding of biosecurity practices and antibiotic usage patterns 

Dear Dr. Zanella:

I'm pleased to inform you that your manuscript has been deemed suitable for publication in PLOS ONE. Congratulations! Your manuscript is now with our production department. 

Kind regards, 

on behalf of

Dr. Iddya Karunasagar 

Academic Editor

PLOS ONE